# Curvature-Modulated Si Spherical Cap-Like Structure Fabricated by Multistep Ring Edge Etching

**DOI:** 10.3390/mi11080764

**Published:** 2020-08-10

**Authors:** Tieying Ma, Jiachen Wang, Dabo Li

**Affiliations:** College of Optical and Electronic Technology, China Jiliang University, Hangzhou 310013, China; wjc857990645@163.com (J.W.); li_dabo@dahuatech.com (D.L.)

**Keywords:** MEMS, silicon, wet etching, TMAH

## Abstract

To create approximately spherical structures with curved sidewalls, this paper presents a method for building a series of decreasing slopes along the sidewall of a circular truncated cone. The multistep ring-edge etching technology of first reducing the concentric mask and then cutting the top off to create a mesa shape can be used to form the slopes. This wet-etching method avoids the constraints of crystallographic properties with surfactant-added Tetramethylammonium hydroxide (TMAH), enabling the manufacture of successive given inclination angles, the precise modulation of the spherical curvature by reduction design of concentric masks, and the setting of etching time. The newly approximated spherical Si microstructure patterns can be used for microlenses, quartz crystal resonators, micropulleys, and other applications. The present research is an approach to fabricate advanced microelectromechanical systems (MEMS) curved-surface structures, extending the range of 3D structures fabricated by silicon wet etching.

## 1. Introduction

Silicon is the most used material in bulk microstructures with applications in microelectromechanical systems (MEMS). Microstructures based on silicon substrates are, therefore, a key issue in bulk micromachining technology. Bulk Si three-dimensional (3D) micromachined structures (e.g., cavities, grooves, through holes, and mesas) have been an area of extensive research in microelectromechanical systems (MEMS) [1,2,3,4]. From the fabrication point of view, etching techniques are divided into two main categories: deep reactive ion etching (DRIE) and anisotropy wet etching. Due to the etch rate being almost independent of the crystallographic orientation, DRIE can be utilized for etching arbitrarily shaped mask patterns to fabricate 3D structures with a high aspect radio and vertical sidewalls [5,6,7,8]. The micromechanical structures formed by DRIE have nonetheless been limited to those with vertical sidewalls such as a comb structure for pressure sensors and grating for optic sensors. DRIE cannot be adopted for structures with slanted or curved sides. The anisotropic wet chemical etching of silicon plays a significant role in bulk micromachining for the formation of uniquely shaped 3D microstructures. In wet bulk micromachining using anisotropic etchants such as potassium hydroxide (KOH) and Tetramethylammonium hydroxide (TMAH), the crystallographic properties restrict the size and shape of the silicon microstructures [9]. For instance, formations of square cavities, V-grooves, and rock-shaped probes, created by undercutting, have restricted-shape sidewalls corresponding to specific crystallographic planes [10]. Some 3D structures with curved surfaces like crankshafts and pulleys cannot be realized by the above etching methods.

Wafer manufacturing technologies commonly focus on silicon wafers with three principle orientations, namely {111}, {110}, and {100}. Among these three orientations, {100} silicon wafers are the most widely employed for the fabrication of microelectromechanical systems (MEMS) and complementary metal oxide semiconductor (CMOS) devices. In order to extend the range of curved 3D structures in {100} Si, several techniques such as grayscale mask lithography and DRIE [11,12] that can adjust the photoresist profile to be a spherical shape by reflow [13] and two-step anisotropic wet etching [14] have been reported. Grayscale lithography utilizes locally modulated exposure doses by changing glass mask transmission or projection lithography to induce diffraction. Differential exposure doses lead to multiple depths of exposed photoresist across the surface. In dry anisotropic etching, the structure is transferred to the silicon substrate to a specified depth corresponding to the height of the desired final structure. In the grayscale method, it is very difficult to fabricate a shape with precise modulation because of the limited range of etch selectivity (normally 0.3–2), nonuniformity, and surface roughness involved in following DRIE [13]. In a reflow-photoresist system, the height of the photoresist shape is modulated by varying the spin-coating rotation speed and the viscosity of the photoresist. These technologies, however, are restricted to a limited range of shapes or do not utilize batch processes. Nevertheless, the imprecise modulation of photoresist or the limited range of etch selectivity makes it difficult to control the shape and curvature of structures. In two-step KOH etching, however, a curved shape can be determined by three parameters: the first etching depth, the second etching depth, and the mask offset between the first and second etching steps. However, its profile is not smooth enough because of the crystallographic properties.

New techniques should be applied to improve the roundness of the 3D structure and modulate its curvature precisely. Recently, various additives have been studied to improve the etching characteristics of a TMAH solution [15,16,17,18,19,20,21,22,23,24]. Among these additives, C_14_H_22_O (C_2_H_4_O)_n_, *n* = 9–10 (Triton-X-100) can slow down the etching rate of the affected surface orientation, which is frequently exploited for new 3D microstructures. In the etching experiments with Triton-X-100, a few groups have found that the etching rate changes significantly as the curvature of the surface varies. They speculate that the morphology of surfactant aggregates strongly depends on the substrate curvature [25,26,27,28]. However, relevant experimental research has not been carried out in depth.

In this paper, we propose and demonstrate fabrication techniques for the realization of curved-surface 3D structures using multistep mask-modified wet cutting. A series of decreasing slopes along the sidewall of the circular truncated cone are built to form curved sidewalls. In order to modulate the curvature of sidewalls accurately, the study focused on the dependence of the inclinations on various etching conditions such as the etching time, modification of concentric mask, etc. The technology can develop many curved-sidewall 3D structures such as a hemisphere, cone, or semicylinder (Figure 1). Moreover, these structures can be utilized in microlenses, quartz crystal resonators, micropulleys, etc., which can extend the functionality of MEMS.

## 2. Design

A section of the approximate spherical structure was designed and shown in Figure 2. To build the model, a special etching process needs to be designed to create a series of decreased asymptotically inclination angles, α, β, γ, etc. along the sidewall from the bottom up, moving gradually from 90° to 0°. In particular, the curved surface will become smoother with a more gradual transition and more inclination changes. Hence, the processes for the realization can be decomposed into two parts: a circular table with a specified inclination is manufactured and then the top convex circular edge is chamfered for several steps under the protection of a trimming mask. Once the inclination of the circular table and the chamfered surface surrounding the top convex circular edge can be modulated precisely by ring-etching, an approximate hemisphere with high-precision curvature surface will be achieved.

In order to construct the given inclinations at different mesa heights, the relationship between height, dip angle, and etching process should be studied. It is reported in the literature that the etching temperature and additive concentration only affect the surface roughness and have little influence on morphology [23,24], so we studied the influence of the etching time on the morphology of the round table. In our experiment, 450 μm thick, four-inch (100) silicon wafers were used for the fabrication of all presented structures. A 50 μm diameter circular SiO_2_ mask was patterned on the (100) Si wafer, and 25 wt % TMAH and 0.1% *v/v* Triton-X-100 were used as the main etchant and surfactant, respectively. Circular mesas were fabricated at 70 °C for 40 min, 50 min, and 60 min, respectively (Figure 3). The mesa’s height and inclination degree were measured and calculated by a step profiler (Table 1). As can be seen, not limited by crystallographic properties, the inclination degree significantly reduces in response to the etching time. We also found that for the first 15 min of corrosion, the etching rate was extremely slow, and after 15 min, the etching rate increased and approached a constant value of 0.4 μm/min. According to the analysis, in the initial etching, Triton-X-100 attaches to the surface to form a protective layer that TMAH should go through by diffusion. After 15 min, the diffusion and chemical reaction tend to balance, while the etching rate remains constant. This provides a reference for our subsequent experiments.

After realizing a given inclination table, the wafer is now ready for the next etching step, aiming at the formation of a series of decreasing inclination angles around the top circular edge of the table. Since a high undercutting rate is desirable along the convex edge, if not restricted by Si crystallographic properties, the design of successive reduced-size masks and the etching of TMAH-added Triton-X-100 is suitable for this purpose, as mentioned before. In particular, different sizes of concentric circular masks were designed; the smaller mask must be aligned on the bigger one each time to remove the ring mask (Figure 4a). Then, the convex corner along the top edge of the mesa is cut (Figure 4b). The newly formed dip angle is smaller than that formed in the previous etching step. Moreover, the slope of chamfering can be modulated by the reduction between the diameters of the two concentric masks. After cutting the top corners several times, an approximate spherical structure with inclinations decreasing gradually from bottom to top can be achieved. It is worth emphasizing that the curvature of sidewalls can be modified precisely by etching time and reduction of the mask.

## 3. Experimental Procedure

In this case, a two-step etching method was applied to cut around the top edge of the round table and form a “ger” model, roughly similar to the spherical structure. A round table was formed, the height of which was about half the diameter of the bottom surface. In previous experiments, etching for 60 min could result in a round table of about 18 μm in height. Therefore, we used a mask with a 30 μm diameter to prepare the round table, so that the aspect ratio of the structure formed was close to that of the spherical structure. The fabrication steps for the realization of the two-step etching are summarized below.

For a single (100) Si wafer, the oxide layer was patterned to form a 30 μm diameter circular mask by photolithography, followed by oxide film etching in buffered hydrofluoric acid (BHF) (Figure 5b). Thereafter, the first step of the etching was performed in wet etchant TMAH + Triton at 70 °C for 60 min to form a circular table, as shown in Figure 5c. Then, it was immersed in BHF for about 1 min in order to remove the rest of the oxide layer mask (Figure 5d). After removing the oxide masking layer, a second etching was performed again for 10 min to chamfer the top edge (Figure 5e). The shapes of the fabricated structures are shown in Figure 6.

As can be seen above, the “ger” model is more similar to a spherical structure than a circular mesa because this new model includes two inclinations: the original one around the bottom ring sidewall and the decreased one around the top edge. It takes only 10 min to cut the convex top edge of the circular table. In this time range, the etching rate to the bottom circular table was extremely small, so the bottom dip angle and the depth of the circular table were almost unchanged. When the convex top edge was cut, the morphology of other parts was basically unchanged, which is helpful for the construction of a spherical structure. Neither of the two degrees were restricted by crystallographic properties, so they could be modulated by the processing conditions such as the etching time and reduction of mask. The proposed spherical structure can be realized depending on the finer indenting of the concentric mask and the refinement of the alignment technology. Therefore, according to the desired shape of the etched profile, the following processes can be adjusted in two ways: increasing the etching steps and modulating the etching time for every step.

Three-step etching was adopted to make further improvements to the spherical structure. After the first etching to form a table, the circular mask is partly removed around the edge and a second etching occurs. Finally, the whole mask is removed to implement the third cutting. The scale parameters of the obtained spherical crown microstructure can be designed according to the experimental results in Table 1 (40 min etching time). It can be calculated that the underside diameter of a crown 12.3 μm in height and with a dip angle of 46.5° was about 57.2 μm (Figure 7). Considering that the diameter of the round table mask should be slightly smaller than that of the ground under the round table, a round table mask was designed at 50 μm. In order to build two equally reduced inclination angles from bottom to top, the diameter of the mask is reduced by about half (30 μm), and the time of the two cutting angles should be close to each other (15 min and 10 min).

The schematic process steps are illustrated in Figure 8. The wafers were thermally oxidized in order to grow a mask (Figure 8a). The oxide layer was then patterned to a 50 μm diameter circular mask by photolithography, followed by oxide etching in BHF (Figure 8b). Thereafter, the first etching was carried out in TMAH + Triton for 40 min to define the shape of the mesa, as shown in Figure 8c. Aligned with a 30 μm diameter concentric circular mask, the oxide layer was partly removed along the edge of the circular mask and decreased (Figure 8d). Again, the wafer was dipped into TMAH + Triton for 15 min to the cut edge’s angle (Figure 8e). After the second etching, the wafer was put into BHF to remove the entire mask, as shown in Figure 8f, and the wafer was now ready for the third etching step for 10 min, aiming at a high chamfering rate of the top (Figure 8g). Breaking the crystallographic restrictions, the final Si spherical structure can be achieved, with continuous decreased inclinations along the sidewall. Figure 9 shows the scanning electron microscope (SEM) picture of promising structures with perfectly curved sidewalls and a smooth etched surface finish. Similarly, our results for three-step etching compared remarkably well with those from two-step etching due to one more mask being aligned.

## 4. Results and Discussion

To quantitatively investigate the similarity of the structure to a standard spherical cap-like structure, particularly in terms of the uniformity between different scanned paths, profiler measurements were carried out with a latitude ranging *θ* from 0 to π; thereafter, we rotated the structure on its axis at π/3 and 2π/3, respectively, and remeasured it point by point (Figure 10). Figure 11 shows an example of the surface profile of the fabricated structure represented by three step height curves (*H*) in relation to the scanned path (*L*). From this figure, the proposed method makes it possible to fabricate a structure with a precisely modulated surface shape. The heights of the spherical cap-like structure can be controlled to the desired value by checking the etching time of the wet chemical process. The variation of inclinations on the spherical cap-like structure’s surface can be controlled by the etching time and the reduction of the mask. On the whole, the scanned paths (as seen from the profile) seemed to correspond well with the expected shape.

The radius of curvature ρ is determined from the measured step height using the following equation:(1)$$\rho =\frac{{\left(1+H^{\prime 2}\right)}^{3/2}}{\left|H^{\prime \prime }\right|},$$
where *H* is the scanned step height. As illustrated in Figure 12, the variation tendency of the radius of curvature for the three different scanned paths was almost the same: the radius of curvature located at the edge of structure (*θ* = 0, π) exhibited the maximum values, which decreased slightly toward the center, especially when the region near the center (*θ* = π/2) reached a minimum. These differences were small and can probably be attributed to small deviations in the etching conditions. The statistics were extracted and analyzed by Origin 8.0 (Table 2). It is speculated that the sawtooth interference on the curves originates from the roughness of the surface, which can be improved by optimizing the etching process parameters such as the etching temperature, additive concentration, etc. Theoretically, the trend of scanning paths vs. the radius of curvature should present a horizontal line on a standard spherical cap-like structure. Hence, the slight U-shaped tendency implies that these decreased inclination angles should be better modulated, which can be done by the size design of the concentric circle masks and etching time.

Now, the question is, why does the inclination change along with the etching time? The proposed cutting model answers this question on the basis of the fact that additives are not evenly adsorbed at different parts of the sidewall. As reported in [29,30], the hydrophobic surface is more likely to absorb the surfactant molecules. The Si (100) wafer under the mesa is hydrophobic for its H-terminations, while the SiO_2_ mask above the mesa is hydrophilic due to the hydroxyl groups, resulting in a gradual distribution of additives from dense to sparse, moving up the sidewall. Moreover, the thicker additive layer protects the bottom sidewall from the faster etching than the top; a significant gradient etching from top to bottom will form an inclination, which can be modulated by the etching time. After removing a concentric ring SiO_2_ mask, a convex edge of the Si circular mesa appeared at the intersecting surface. The next question is why etching can be observed at the convex edge, but not at the concave edge. After removing the mask, concave and convex circular edges formed by the intersection of two Si (100) surfaces and the sidewall had a similar absorption of surfactant molecules. According to the dangling bond model [13], we can easily notice the difference in the bond structure at the two intersecting edges. Every silicon atom belonging to the convex edge consists of one dangling bond (Figure 13b), while the atoms belonging to the concave edge do not contain any dangling bonds (i.e., all the bonds are engaged). The absence of the dangling bond at the concave edge suppresses etching. Hence, the chamfering occurs along the circular convex edge.

## 5. Conclusions

New multistep ring-edge etching methods for the realization of a precise curvature-modulated Si spherical cap-like structure with a series of decreasing slopes were proposed and successfully demonstrated. Since the additive Triton-X-100 absorbed on the surface of the bulk silicon can inhibit the anisotropic corrosion of the etchant (TMAH), the lateral crystal silicon can break through the restriction of crystallographic properties, making it possible to construct any given dip angle. The spherical cap-like structure, on the other hand, can be viewed as a combination of a series of circular truncated cones with decreased inclination. Hence, fabrication methods are developed using a successive reduction of concentric masks and TMAH wet etchant added with Triton-X-100 to build spherical cap-like structures with a series of given decreased inclination angles along the sidewall. The curvature of its side can also be modulated by adjusting the mask reduction and etching time to obtain ellipsoids with different radians.

In future experiments, the smoothness of the curved structure can be further improved by optimizing the etching time and mask reducing parameters and increasing the number of side cuts. It is worth noting that, compared with the previously reported methods, this technique is not restricted by crystallographic properties, photoresist characteristics, or etch selectivity in DRIE, so it has greater space for improvement in the shape of the precisely modulated spherical cap-like structure.

In a further study, using a similar technique, we can successively side-cut a semicylinder by a series of coaxial rectangular masks with unchanged length and gradually reduced width; we can even form cones by a circular mask. Based on technologies of curved-sidewall 3D structures, the present research has a wide scope including, but not limited to, the fabrication of microlenses, quartz crystal resonators, micropulleys, etc. The present research expands the range of curved-surface 3D structures fabricated by silicon anisotropic etching.

## Figures and Tables

**Figure 1 micromachines-11-00764-f001:**
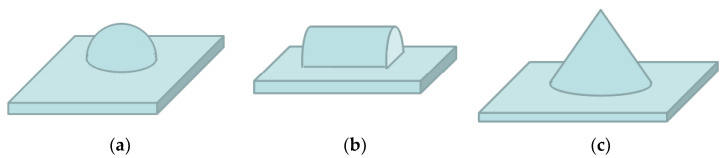
Shape of 3D microstructures with a sidewall of approximately curved surfaces: (**a**) hemisphere; (**b**) semicylinder; (**c**) cone.

**Figure 2 micromachines-11-00764-f002:**
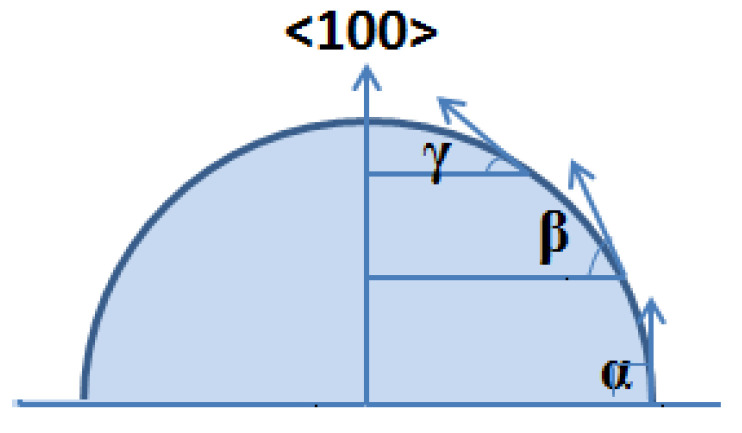
Section of approximate hemisphere.

**Figure 3 micromachines-11-00764-f003:**
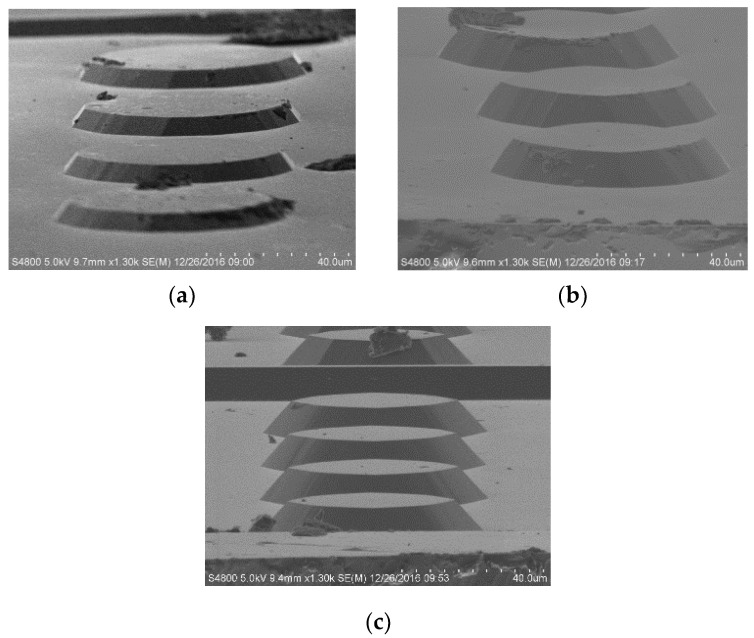
Shape of circular mesas in Tetramethylammonium hydroxide (TMAH) + Triton-X-100 solution after different etching times: (**a**) 40 min; (**b**) 50 min; (**c**) 60 min.

**Figure 4 micromachines-11-00764-f004:**
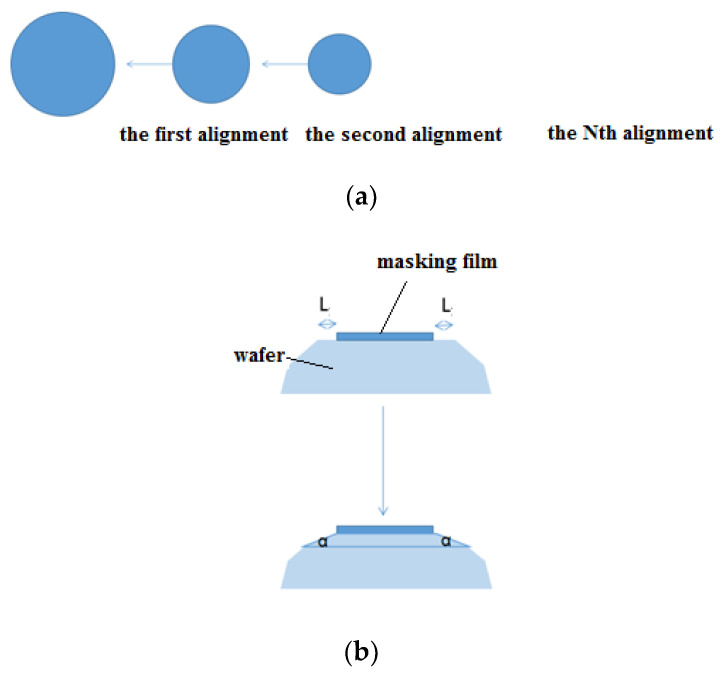
Schematic view of the process steps: (**a**) a circular mask with smaller diameter is aligned to a larger one successively to realize the approximate hemisphere model; (**b**) a circular truncated cone with perfect cutting edge angle on the top profile.

**Figure 5 micromachines-11-00764-f005:**
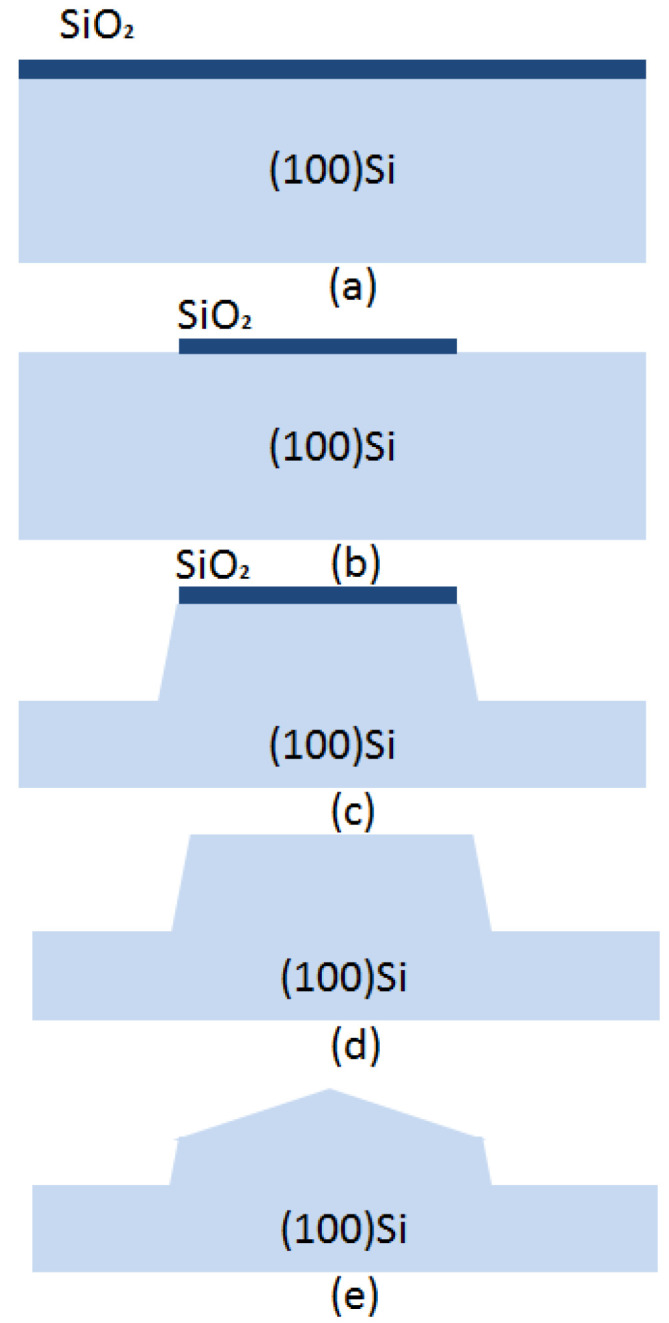
Two-step etching process. (**a**) a single(100) Si wafer with oxide layer; (**b**) the oxide mask patterned with a 30 μm diameter circular mask; (**c**) a circular table was formed by wet etching; (**d**) the rest of the oxide layer mask was removed; (**e**) a second etching was performed to form final shape.

**Figure 6 micromachines-11-00764-f006:**
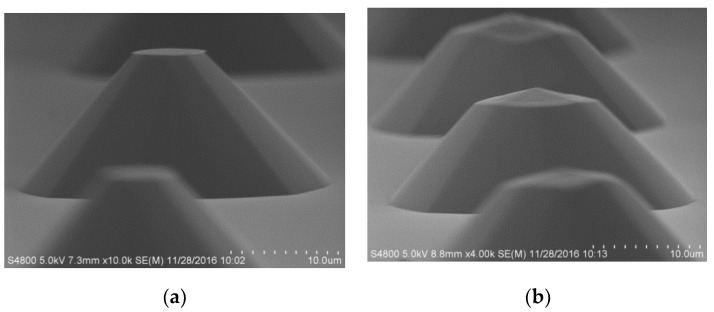
Two-step etching scanning electron microscope (SEM) of “ger” model: (**a**) etching is performed in 25 wt % TMAH + Triton at 70 °C for 60 min to form a circular mesa; (**b**) the “ger” model was fabricated by chamfering the top edge.

**Figure 7 micromachines-11-00764-f007:**
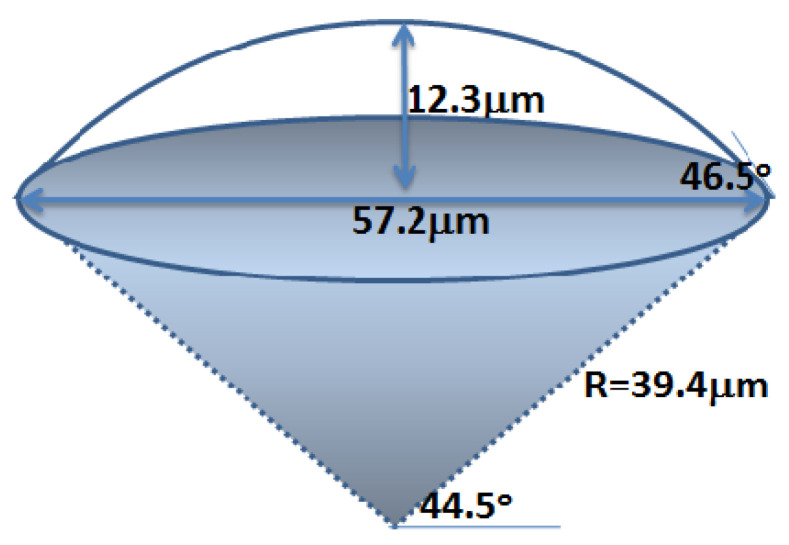
Parameter designation of the spherical crown microstructure.

**Figure 8 micromachines-11-00764-f008:**
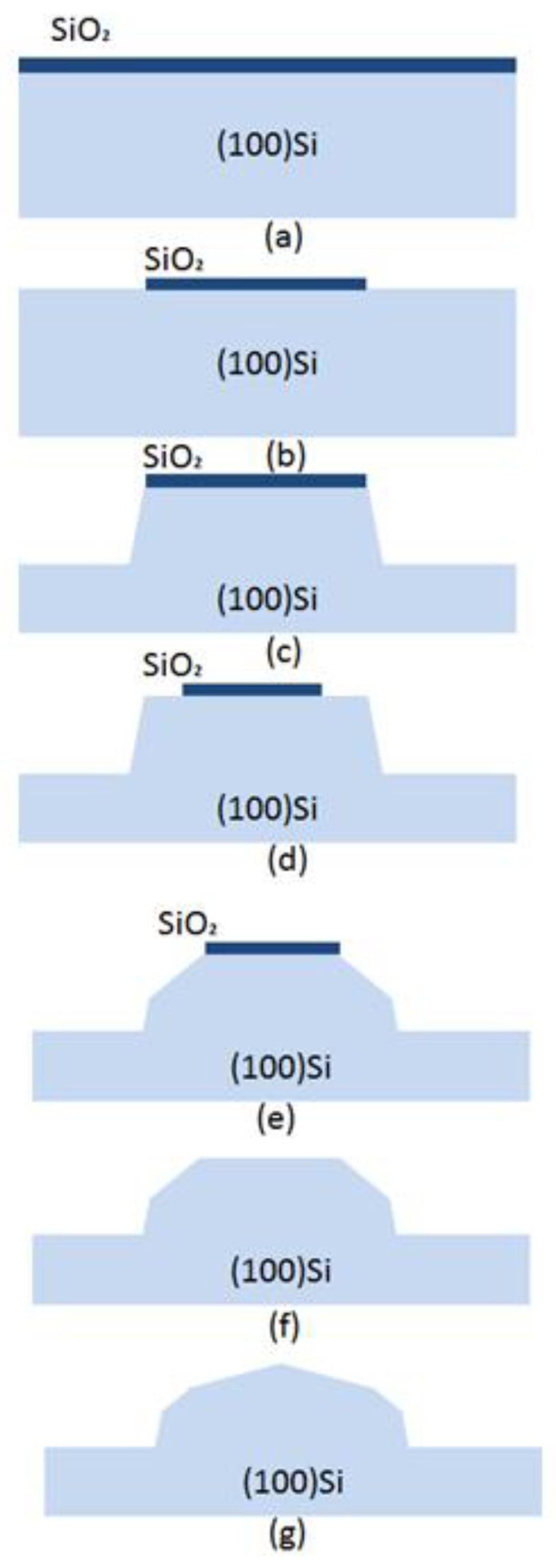
Three-step etching processing. (**a**) Oxide layer was grown on a single(100) Si; (**b**) The oxide layer was patterned to a 50 μm diameter circular mask; (**c**) a circular table was formed by wet etching; (**d**) the oxide layer was partly removed to form a 30 μm diameter concentric circular mask; (**e**) the wafer was cut to form edge’s angle; (**f**) the rest of the oxide layer mask was removed; (**g**) a third etching was performed to form final shape.

**Figure 9 micromachines-11-00764-f009:**
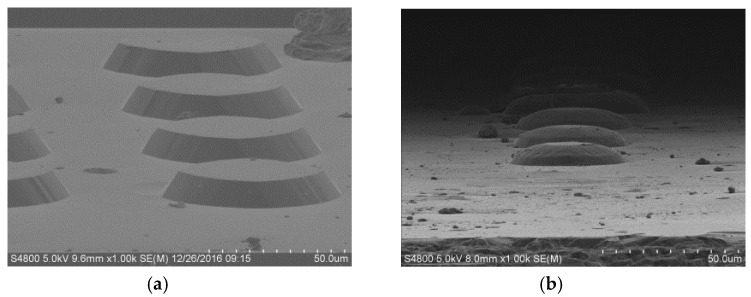
Three-step etching SEM of the spherical cap-like structure. (**a**) The first etching was patterned by a 50 μm diameter circular mask to form a circular truncated cone; (**b**) the second etching was performed by a 30 μm diameter mask to chamfer the top edge’s angle; (**c**) the mask was removed at the third etching and a spherical cap-like structure is formed.

**Figure 10 micromachines-11-00764-f010:**
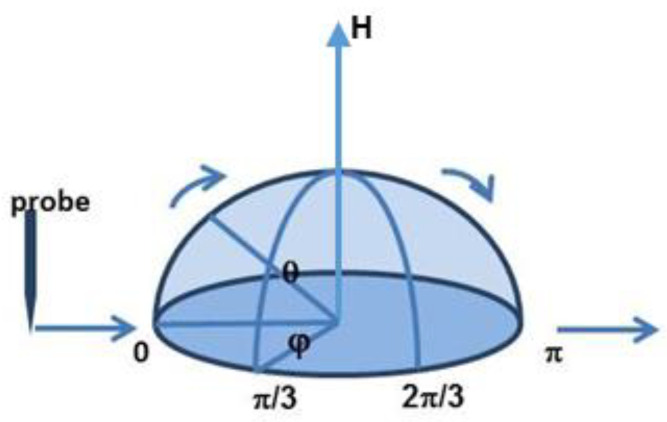
Scanned path by step profiler with a latitude ranging *θ* from 0 to π and longitude *φ* = 0, π/3, and 2π/3, respectively.

**Figure 11 micromachines-11-00764-f011:**
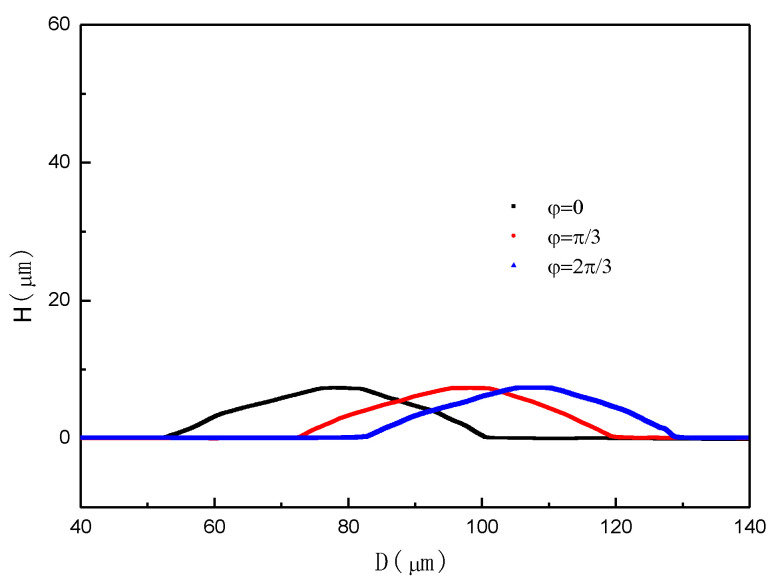
Measurements of three step height curves (*H*) in relation to the scanned path (*L*) with longitude *φ* = 0, π/3, and 2π/3, respectively.

**Figure 12 micromachines-11-00764-f012:**
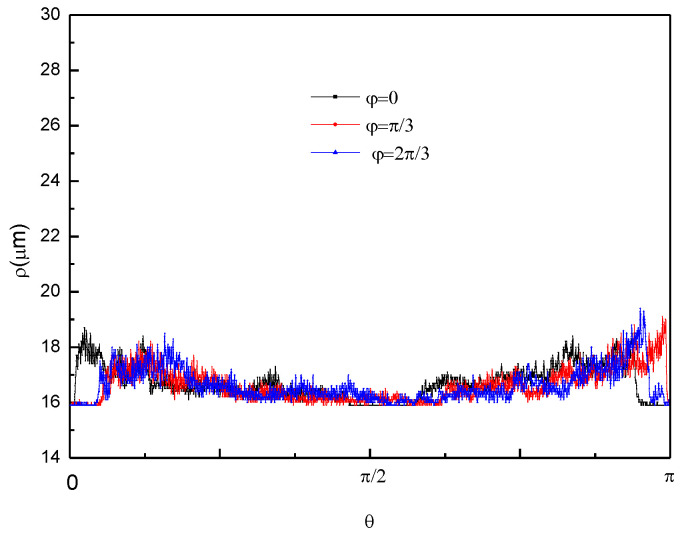
Radius of curvature (*ρ*) in relation to the scanned path (*L*) with longitude *φ* = 0, π/3, and 2π/3, respectively.

**Figure 13 micromachines-11-00764-f013:**
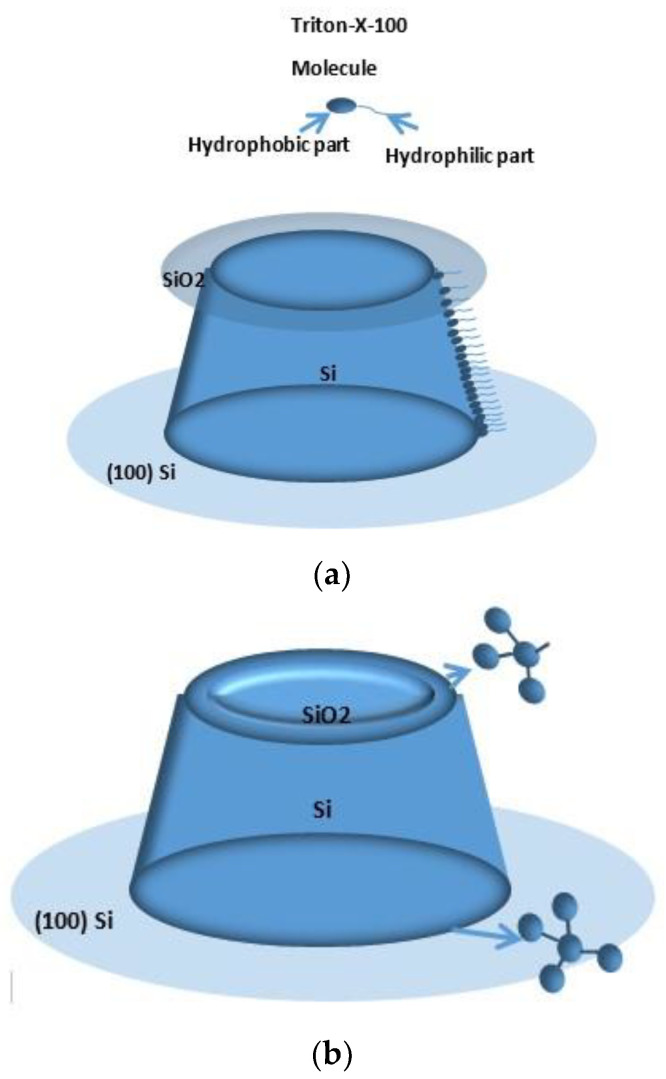
Model explaining the wet etching on the silicon surface: (**a**) additive’s gradient absorption model on sidewall; (**b**) dangling bond model along circular convex edge.

**Table 1 micromachines-11-00764-t001:** Shape parameters of the circular mesas etched in Tetramethylammonium hydroxide (TMAH) + Triton-X-100 solution after different times.

Etching Time (min)	Mesa Height (μm)	Inclination Degree of Sidewall (°)
40	12.3	46.5
50	14.15	44
60	18.16	40

**Table 2 micromachines-11-00764-t002:** Statistics (mean, standard deviation, minimum, median, and maximum) for the radius of curvature, calculated by Origin 8.0.

*φ*	Mean (μm)	Standard Deviation	Minimum (μm)	Median (μm)	Maximum (μm)
0	16.75	0.59	15.9	16.7	18.7
π/3	16.65	0.57	15.9	16.5	19.1
2π/3	16.71	0.65	15.9	16.5	19.4

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
