# Peer review of "Curvature-Modulated Si Spherical Cap-Like Structure Fabricated by Multistep Ring Edge Etching"

_micromachines, 2020, doi:10.3390/mi11080764_

Round 1
Reviewer 1 Report
The revised paper can be published as is.
Author Response
Dear Reviewer:
Thank you very much for your careful review and constructive suggestions with regard to our manuscript. Those comments are all valuable and very helpful for revising and improving our paper, as well as the importance guiding significance to our researches.
Reviewer 2 Report
To my understanding, the proposed multi-step wet-etching process yields spherical-cap-like structures with piece-wise linear side-wall profiles but not a properly curved one. Furthermore, the manufactured structures are far from being hemispherical. From this perspective, I think both the title and the abstract is misleading. The advantages of your method over grayscale lithography is not clear, and the entire development process is not supported by a string motivation. Still, while the method might be some interest for the readers, in its current form the manuscript is very difficult to read due to the use of English language. In order to be able to assess the merits of the work, the manuscript needs to go through and extensive editing step. Once this is done, a peer review can adequately be performed. I therefore cannot recommend the manuscript for publication.
Author Response
Dear Reviewer:
Thank you very much for your careful review and constructive suggestions with regard to our manuscript. Those comments are all valuable and very helpful for revising and improving our paper, as well as the importance guiding significance to our researches. We have studied comments carefully and have made correction which we hope meet with approval. The main corrections in the paper and the responds to the reviewer's comments are as attachment.

Reviewer 3 Report
This paper expands on an experiment designed to fabricate curved structures using a combination of TMAH and surfactant which, combined to successive etches, lead to the final design.
The paper is interesting and deserves to be considered for publication. However, there is a need for extensive English edits as this detracts from the overall reading experience. I have several additional comments that should be taken care of before possible publication:
- Introduction: references 6 to 8 do not seem very relevant, except for self-referencing.
- How is the alignment of the subsequent masks performed? What is the required precision on those?
- SEMs presented would in general benefit from higher magnification.
- The y scale on Fig. 10 should be modified.
Author Response
Dear Reviewer:
Thank you very much for your careful review and constructive suggestions with regard to our manuscript. Those comments are all valuable and very helpful for revising and improving our paper, as well as the importance guiding significance to our researches. We have studied comments carefully and have made correction which we hope meet with approval. The main corrections in the paper and the responds to the reviewer's comments are as attachment.

This manuscript is a resubmission of an earlier submission. The following is a list of the peer review reports and author responses from that submission.
Round 1
Reviewer 1 Report
This paper describes a method to fabricate curved-surface 3D structures on a silicon wafer using multi-step wet chemical etching. The authors demonstrate the realization of a series of decreasing slopes along the sidewall of a circular truncated cone, forming curved sidewalls. The paper is interesting and can be published after considering the following remarks:
1. The wafer size and thickness used should be mentioned
2. It should be mentioned explicitly in the text figure caption that the shown images are scanning electron microscopy (SEM) images. A marker should be added in the images to indicate the dimension scale.
3. Higher magnification SEM images should be added to show the surface morphology of the circular truncated cone.
4. In references 9 and 12 the publication year should be added. Some other references are incomplete and need to be completed.
5. Few grammatical errors exist and need to be corrected.
Author Response
Dear reviewer, thanks for your review. In response to the review comments, I have carefully revised and replied as follows:
- The wafer size and thickness used should be mentioned
A: A sentence has been added in paper: In our experiment, 450mm-thick Four-inch (100) silicon wafers are used for the fabrication of all presented structures.
- It should be mentioned explicitly in the text figure caption that the shown images are scanning electron microscopy (SEM) images. A marker should be added in the images to indicate the dimension scale.
A: Scale bars have been added.
- Higher magnification SEM images should be added to show the surface morphology of the circular truncated cone.
A: I’m sorry that the experiment has been over for a long time. It’s difficult to provide new higher magnification SEM images.
- In references 9 and 12 the publication year should be added. Some other references are incomplete and need to be completed.
A: The publication years of reference 9 and 12 have been added. Other references have been checked again.
- Few grammatical errors exist and need to be corrected.
A: Grammatical errors have been checked and revised again.
Reviewer 2 Report
The authors report on a method to create 3D structures on silicon. Although the overall motivation and need for such techniques is clear I'm sorry I I cannot accept the paper in its current form. THis is due to many aspects, some are listed below:
The authors should at first remove the word nove. Their work appears original but they combine known techniques to novel appear inappropriate.
How can you make quatz elements and micro lenses in silicon?
The show some sketches of unique structures can can be potentiall made in Fig. 1 but do not provide a clear strategy how to get there.
The sketch in Fig. 2 and 4 are not supportive - I have doubts that I understood the method correctly and I don'T feel confident to redo experiments. But this must be the level of a high-quality paper.
The technological strategy is hardly explained and its not clear how the multiple masks are applied on a prepatterned surface?
The SEM images are not labeled. Missing scale bars!
The research might be interesting and therefore I would encourage to resubmit but the entire story is not well told so far. So ensure a clear presentation of you arguments and findings, provide all experimental details, create at the beginning of you paper a detailed schematic that illustrates your approach clearly.
Author Response
Dear reviewer, I’m very sorry not to present my findings and details clearly. As a non-native English writer, it’s not easy to illustrate my approach in depth. I have tried my best to check the grammar error for several times. In response to your questions, please see the attachment

Round 2
Reviewer 2 Report
I‘m sorry but the changes are insufficient. Therefore, I have to vote for paper rejection.